# Fast Convergence of Belief Propagation to Global Optima: Beyond Correlation Decay

**Frederic Koehler**
Department of Mathematics
Massachusetts Institute of Technology
Cambridge, MA 02141
fkoehler@mit.edu

## Abstract

Belief propagation is a fundamental message-passing algorithm for probabilistic reasoning and inference in graphical models. While it is known to be exact on trees, in most applications belief propagation is run on graphs with cycles. Understanding the behavior of "loopy" belief propagation has been a major challenge for researchers in machine learning and other fields, and positive convergence results for BP are known under strong assumptions which imply the underlying graphical model exhibits decay of correlations. We show, building on previous work of Dembo and Montanari, that under a natural initialization BP converges quickly to the global optimum of the Bethe free energy for Ising models on arbitrary graphs, as long as the Ising model is *ferromagnetic* (i.e. neighbors prefer to be aligned). This holds even though such models can exhibit long range correlations and may have multiple suboptimal BP fixed points. We also show an analogous result for iterating the (naive) mean-field equations; perhaps surprisingly, both results are dimension-free in the sense that a constant number of iterations already provides a good estimate to the Bethe/mean-field free energy.

## 1  Introduction

Undirected graphical models, also known as Markov Random Fields, are a general, powerful, and popular framework for modeling and reasoning about high dimensional distributions. These models explain the dependency structure of a probability distribution in terms of interactions along the edges of a (hyper-) graph, which gives rise to a factorization of the joint probability distribution, and the absence of edges in this graph encodes conditional independence relations.

Ising models are a special class of graphical models with a long history and which are popular in applications; they model the interaction of random variables valued in a binary alphabet ($\{\pm 1\}$) with exclusively pairwise interactions. Explicitly, the joint pmf of an Ising model is

$$\Pr(X = x) = \exp\left(\frac{1}{2}x^T J x + h \cdot x - \log Z\right) \tag{1}$$

where $x \in \{\pm 1\}^n$, $J : n \times n$ is an arbitrary matrix describing the pairwise interactions between nodes (with zero diagonal), $h$ is an arbitrary vector encoding node biases, and $Z$ is a proportionality constant called the *partition function*. Historically, Ising models originated in the statistical physics community as a way to model and study phase transition phenomena in magnetic materials; since then, they have attracted significant interest in the machine learning community and have been applied in a wide variety of domains including finance, social networks, neuroscience, and computer vision (see e.g. references in [18, 23, 30, 11].

Performing sampling and inference on an Ising model is a major computational challenge for which a wide variety of approaches have been developed. One family of methods are Markov-Chain Monte Carlo (MCMC) algorithms, the most popular of which is Gibbs sampling (also known as Glauber dynamics) which resamples the spin of one node at a time from its conditional distribution. When run sufficiently long, MCMC methods will draw samples from the true distribution (1); unfortunately, it is well-known in both theory and practice that MCMC methods may become stuck when the probability distribution (1) exhibits multi-modal structure; for example, on an $n \times n$ square lattice the Glauber dynamics required exponential time to mix in the low temperature phase [20].

A popular alternative to markov chain methods are *variational methods*, which typically make some approximation to the distribution (1) but often run much faster than MCMC. These methods usually reduce inference on the Ising model to some (typically non-convex) optimization problem, which is solved either by standard optimization methods (e.g. gradient ascent) or by more specialized methods like *message-passing algorithms* (e.g. Belief Propagation). Because of the non-convexity, these methods are typically not guaranteed to return global optimizers of the corresponding variational problem. Indeed, these optimization problems are NP-hard to approximate for general Ising models (see e.g. [12] for the case of mean-field approximation).

*Belief propagation* (BP) is a celebrated message passing algorithm which is known to be closely related to the Bethe approximation. It is a fundamental algorithm for probabilistic inference [25] which plays a fundamental role in a variety of applications like phylogenetic reconstruction, coding, constraint satisfaction problems, and community detection in the stochastic block model (see e.g. [21, 23, 6]); it is also closely connected to the "cavity method" in statistical physics [21]. Although BP is observed to works well for many problems, there are few settings on general graphs (i.e. with loops) where it provably works. For example, BP with random initialization is conjectured to achieve optimal reconstruction in the 2-community SBM [6] but no rigorous proof of this result is known.

In this work, we consider two popular variational approximations, the naive mean-field approximation and the Bethe approximation to the Ising model, and the corresponding heuristic message-passing algorithms which are usually used to solve these optimization problems: mean-field iteration and belief propagation. We show that under a natural and popular assumption of *ferromagneticity* (that $J_{ij} \geq 0$ and $h$ has consistent signs; a.k.a. as an *attractive* graphical model) that these methods do indeed converge to global optimizers of their optimization problems, under a natural initialization, and moreover that their convergence rate is fast and dimension-free in the appropriate sense.

## 1.1   Background: Variational methods and belief propagation

We can describe the variational methods we consider in terms of optimization problems whose goal is to estimate $\Phi := \log Z$, the log partition function or *free energy* of the Ising model. This is natural to consider because other important quantities can be recovered by differentiating $\log Z$ in some parameter, and because the ability to construct sufficiently precise estimates for $Z$ is equivalent to approximate sampling for any self-reducible family of models [15]. We note throughout this section we specialize to Ising models, but all of these notions generalize straightforwardly to general Markov random fields (a.k.a. factor models) — see [21] for a more detailed discussion.

The starting point for these variational methods is the *Gibbs variational principle* [21] which states

$$\log Z = \max_{P \in \mathcal{P}(\{\pm 1\}^n)} \mathbb{E}_P[\frac{1}{2}x^T J x + h \cdot x] + H_P(X) \tag{2}$$

where $P$ ranges over probability distributions on $\{\pm 1\}^n$ and $H_P(X)$ is the entropy of $X$ under $P$. This formula is derived by observing the Gibbs measure minimizes the KL divergence to itself and expanding.

The *(naive) mean-field approximation* is given by restricting (2) to product distributions and finding the maximum of the functional

$$\Phi_{MF}(x) := \frac{1}{2}x^T J x + h \cdot x + \sum_i H\left(Ber\left(\frac{1+x_i}{2}\right)\right) \tag{3}$$

where $H(Ber(p)) = -p \log p - (1-p) \log(1-p)$ is the entropy of a Bernoulli random variable. Information-theoretically, the optimizer(s) $x^*$ of this optimization problem corresponds to the

marginals of a product distribution $\nu$ which minimizes the KL-divergence from the Gibbs measure $\mu$ (defined by (1)) to $\nu$. Note that this always gives a lower bound on $\log Z$.

By considering the first-order optimality conditions for (3), one arises at the *mean-field equations*

$$x = \tanh^{\otimes n}(J \cdot x + h) \tag{4}$$

where $\tanh^{\otimes n}$ denotes entry-wise application of $\tanh$. The *mean-field iteration* is the natural iterative algorithm which starts with some $x_0$ and applies (4) iteratively to search for a fixed point. The error of the mean-field approximation has been extensively studied; the approximation is guaranteed to be accurate when $\|J\|_F = o(n)$ (informally, in unfrustrated models with large average degree); see e.g. [2, 13, 1]. For example, the recent result of [1] shows that $|\log Z - \max_x \Phi_{MF}(x)| = O(\sqrt{n}\|J\|_F)$ and the result of [10] gives even better bounds for some models.

The naive mean-field approximation can be inaccurate on very sparse graphs; the *Bethe approximation* is a more sophisticated approach which has the benefit of being exact on trees [21], and which is always at least as accurate as the mean-field approximation in ferromagnetic models [26]. The Bethe free energy is the maximum of the (typically non-convex) functional

$$\Phi_{Bethe}(P) := \sum_{i \sim j} J_{ij}\mathbb{E}_{P_{ij}}[X_iX_j] + \sum_i h_i\mathbb{E}_{P_i}[X_i] + \sum_{i \sim j} H_{P_{ij}}(X_i, X_j) - \sum_i (\deg(i)-1)H_{P_i}(X_i) \tag{5}$$

where $P$ lies in the polytope of locally consistent distributions (equivalently, $SA(2)$ in the Shereli-Adams hierarchy[1]). Explicitly this polytope is given by constraints:

$$\sum_{x_i} P_{ij}(x_i, x_j) = P_j(x_j) \qquad \text{for all } i, j \text{ neighbors}$$

$$\sum_{x_i} P_i(x_i) = 1 \qquad \text{for all } i$$

$$P_i(x_i) \geq 0 \qquad \text{for all } i, x_i$$

One can derive the Bethe-Peierls (BP) equations from the first-order optimality conditions for this optimization problem; this connection is involved and is discussed further in Section 3.1. Just like the mean-field equations, the BP equations can be iterated to search for a fixed point, in which case one recovers the belief propagation updates for this setting. Explicitly, for edge messages $\nu_{i \to j}$ with $\nu_{i \to j} \in [-1, 1]$ the consistency equation is

$$\nu_{i \to j} = \tanh(h_i + \sum_{k \in \partial i \setminus j} \tanh^{-1}(\tanh(J_{ik})\nu_{k \to i})) \tag{6}$$

where $\partial i$ denotes the neighborhood of node $i$. Intuitively, this equation describes the expected marginal of node $X_j$ in the absence of the edge between $i$ and $j$. Given $\nu$ which solves these equations, the BP estimate for $\mathbb{E}[X_i]$ can be written as $\nu_i := \tanh(h_i + \sum_{k \in \partial i} \tanh^{-1}(\tanh(J_{ik})\nu_{k \to i}))$. BP also gives an estimate for the free energy in terms of its messages (see equation (7) in Section 3.1).

The above derivation of ("loopy") belief propagation from the the Bethe free energy for general graphical models is due to [36]. Alternatively, belief propagation can also be derived as the exact solution to computing the partition function of a tree graphical model and this can be found in a variety of places — see e.g. [25].

## 1.2 Our Results

We analyze the behavior of mean-field iteration and belief propagation in ferromagnetic (a.k.a. attractive) models on arbitrary graphs.

**Definition 1.1.** An Ising model is *ferromagnetic* (with consistent field) if $J_{ij} \geq 0$ for all $i$ and $h_i \geq 0$ for every $i$. (We also can allow $h_i \leq 0$ for all $i$, but this is equivalent after flipping signs.)

We show that in ferromagnetic Ising models, belief propagation and mean-field iteration always converge to the true global optimum of the Bethe free energy and mean-field free energy respectively, as long as they start from the all-1s initialization. Moreover we show that these algorithms converge quickly, which makes them fast and practical algorithms for estimating the corresponding objective. We note that these results cannot hold for arbitrary Ising models, as even approximating the mean-field free energy is NP-hard in general Ising models with anti-ferromagnetic interactions [12].

**Theorem 1.2.** *Fix an arbitrary ferromagnetic Ising model parameterized by $J, h$ and let $x^*$ be a global maximizer of $\Phi_{MF}$. Initializing with $x^{(0)} = \vec{1}$ and defining $x^{(1)}, x^{(2)}, \ldots$ by iterating the mean-field equations, we have that[2] for every $t \geq 1$,*

$$0 \leq \Phi_{MF}(x^*) - \Phi_{MF}(x^{(t)}) \leq \min \left\{ \frac{\|J\|_1 + \|h\|_1}{t}, 2 \left( \frac{\|J\|_1 + \|h\|_1}{t} \right)^{4/3} \right\}.$$

**Theorem 1.3.** *Fix an arbitrary ferromagnetic Ising model parameterized by $J, h$, and let $P^*$ be a global maximizer of $\Phi_{Bethe}$. Initializing $\nu_{i \to j}^{(0)} = 1$ for all $i \sim j$ and defining $\nu^{(1)}, \nu^{(2)}, \ldots$ by BP iteration on a graph with $m$ edges we have for every $t \geq 1$,*

$$0 \leq \Phi_{Bethe}(P^*) - \Phi_{Bethe}^*(\nu^{(t)}) \leq \sqrt{\frac{8mn(1 + \|J\|_\infty)}{t}}$$

We also give simple lower bound examples showing these bounds are not too far from optimal; for example, for both algorithms we show the optimal asymptotic rate in $t$ is lower bounded by at least $\Omega(1/t^2)$. We refer to these bounds as *dimension-independent* because under the typical scaling of the entries of $J$ in a ferromagnetic model, they show that mean-field iteration/BP achieve good estimates to the variational free energy density after a constant number of iterations. We explain this more precisely in the next remark:

**Remark 1.4** (Scaling and Dimension-Free Nature). In ferromagnetic models, we usually expect the scaling $\|J\|_1 = \Theta(n), \|h\|_1 = \Theta(n)$ so that all of the terms in the Gibbs variational principle (2) are on the same order, since the entropy scales like $\Theta(n)$ (e.g. the entropy of $Uni(\{\pm 1\}^n)$ is $n \log(2)$). Then the free energy $\log Z$ and its variational approximations all grow like $\Theta(n)$, so when considering the scaling in $n$ one should consider the *free energy density* $\frac{1}{n} \log Z$. Writing the guarantee of Theorem 1.2 for the free energy density when picking the $O(1/t)$ bound, we see $0 \leq \frac{1}{n} \Phi_{MF}(x^*) - \frac{1}{n} \Phi_{MF}(x_t) \leq \frac{\|J\|_1 + \|h\|_1}{nt}$ and the rhs is $\Theta(1/t)$ under the assumption $\|J\|_1 = \Theta(n), \|h\|_1 = \Theta(n)$. We get a similar dimension-free guarantee for BP as long as $m = \Theta(n)$, i.e. the model is sparse, and $\|J\|_\infty = O(1)$ which is a very rarely violated assumption.

**Remark 1.5** (Importance of initialization). The fast convergence rates we show do not hold for other seemingly natural choices of initialization; e.g. if we start BP with initial messages near zero. For a concrete illustration of this, see Figure 2 in the Appendix.

Finally, we build on ideas developed in our analysis of BP to give a different method, based on convex programming, which has worse dependence on $n$ but converges exponentially fast, i.e. can compute the optimal BP solution to error $\epsilon$ in time $poly(n, \log(1/\epsilon))$. This is described in Appendix H; as we explain there, such a method is useful when we care about computing the optimal BP solution accurately in parameter space, as there can be exponentially flat (in terms of $\beta$) directions in the objective.

## 1.3 Related Work

As mentioned above, the general connection between the Bethe free energy and Belief Propagation was established in the work of Yedidia, Freeman and Weiss [36]. They showed that in any graphical model, the fixed points of BP correspond to the critical points of the Bethe Free Energy. However, their theory by itself does not say anything about the behavior of BP when it is not at a fixed point (as is the case during the BP iteration), or which fixed point (if any) it will converge to. In the special case that the edge constraints are relatively weak, e.g. if they satisfy Dobrushin's uniqueness condition [9], one can show that BP converges to a unique fixed point by comparing to what happens

on a tree (see [29, 22]). BP is also known to converge if the graph has at most one cycle [31]. Stronger results are known for BP in Gaussian graphical models, in which case BP can be viewed as an iterative method for solving linear equations [32, 19].

This work builds upon previous work of Dembo and Montanari [7], who studied the convergence of Belief Propagation in ferromagnetic Ising models with a positive external field strictly bounded away from 0. In their analysis they crucially showed that in all such models, BP converges at an asymptotically exponential rate to a unique fixed point if initialized with non-negative messages (other fixed points may exist, but they have at least one negative coordinate). As discussed in [7], this can be thought of as establishing an "average-case" version of correlation decay which goes well beyond the usual "worst-case" setting (which would require there to be a unique global fixed point). From this they derived analytic results for graphs for Ising models which converge locally to (random) trees: these models exhibit an average-case version of *correlation decay*, BP correctly estimates the marginals (e.g. $\mathbb{E}[X_i]$) of the Ising model, and from this derive that the "cavity prediction" for the free energy, which is determined by the tree the graph locally converges to, is correct to leading order. These analytic results were generalized in [8] beyond the Ising case (i.e. to non-binary spins) using some new techniques.

In contrast, in this work we allow for the complete absence of external field, in which case these models may have multiple fixed points, even in the space of nonnegative messages. Furthermore, the optimal BP fixed point often does not correspond to the true marginals of the underlying Ising model[3]. Despite this, we show that BP converges quickly in objective value[4] to the optimal fixed point as long as we start from the all-1s initialization. Another key difference is we are interested in the behavior of BP on general graphs, where the BP estimate cannot always be related to the true free energy; we get around this issue by building on the connection established in [36] to show that the BP result is always equal to the Bethe free energy on any graph, not necessarily locally tree-like.

A very different line of work studies the dense limit of BP in spin glasses and related models, in the form of the TAP approximation and Approximate Message Passing (AMP); for example, see [4, 3]. These results are more concerned with dense models with random edge weights and are motivated by CLT-type considerations; the models they consider are typically far from ferromagnetic and thus require in the dense limit the TAP approximation instead of the naive mean field approximation. Since we consider arbitrary graphs instead of (dense) random graphs, these techniques are not applicable.

Outside of message passing algorithms, we note that in ferromagnetic Ising models it is actually possible to sample efficiently from the Boltzmann distribution by using a special Markov chain which performs non-local updates: this was proved in a landmark work of Jerrum and Sinclair [14]. This result can be used to give an algorithm for approximating the mean-field free energy using a graph blow-up reduction [12]. Also, for ferromagnetic models it was shown previously that the Bethe free energy can be computed in polynomial time using discretization with submodularity-based methods in [16, 33]. The polynomials in the runtime guarantees for both methods are fairly large compared to the message-passsing algorithms discussed in this work.

## 2   Convergence of Mean-Field Iteration

In this section, we give the proof of Theorem 1.2 by analyzing the mean-field iteration. Organizationally, we split this theorem into two (corresponding to the two seperate bounds implied by the min): we prove the first bound in the theorem as Theorem 2.4 and the second $O(1/t^{4/3})$ bound as Theorem 2.6. Omitted proofs are deferred to Appendices A-C corresponding to each subsection.

## 2.1 Main convergence bound

In this section we prove the first $(O(1/t))$ bound appearing in Theorem 1.2, the bound which is better for small $t$; we consider this to be the more significant bound because it gives a meaningful convergence result even when $t = O(1)$ (see Remark 1.4). A key observation in the proof is that the functional $\Phi_{MF}$ is actually concave on a certain subset of the space of product distributions, and that the iteration stays in this region because the iteration is monotone w.r.t. the partial order structure; this allows us to show progress at each step.

For the analysis of mean-field iteration, it will be very helpful to split the updates up into two steps:

$$y^{(t+1)} := Jx^{(t)} + h$$
$$x^{(t+1)} := \tanh^{\otimes n}(y^{(t+1)}).$$

**Lemma 2.1.** *A global maximizer of $\Phi_{MF}$ is in $[0,1]^n$.*

*Proof.* For any $x$, if $|x|$ denotes the coordinate wise absolute value then we observe $\Phi_{MF}(x) \le \Phi_{MF}(|x|)$ since $J, h$ are entrywise nonnegative and the entropy term is preserved. Therefore if $x$ is a global maximizer then so is $|x|$, and by compactness of $[-1, 1]^n$ there exists at least one global maximizer. $\square$

**Lemma 2.2.** *There exists at most one critical point of $\Phi_{MF}$ in $(0,1)^n$.*

Based on these lemmas, we define $x^*$ to be the global maximizer of $\Phi_{MF}$ in $[0,1]^n$. Define $S := \{x \in (0,1]^n : x_i \ge x_i^*\}$.

**Lemma 2.3.** *The mean-field free energy functional $\Phi_{MF}$ is concave on $S$.*

**Theorem 2.4** (Main bound in Theorem 1.2). *Suppose that $x_0 \in S$ and define $(x^{(t)}, y^{(t)})_{t=1}^\infty$ by iterating the mean-field equations. Then for every $t$, $x^{(t)} \in S$. Furthermore*

$$\Phi_{MF}(x^*) - \Phi_{MF}(x^{(t)}) \le \frac{\|J\|_1 + \|h\|_1}{t}.$$

*Proof.* To show that $x^{(t)} \in S$, observe that the mean-field iteration is monotone: if $x \le x'$, then $\tanh^{\otimes n}(Jx + h) \le \tanh^{\otimes n}(Jx' + h)$. Therefore, because $x^* \le x_0$ we see that $x^* = \tanh^{\otimes n}(Jx^* + h) \le \tanh^{\otimes n}(Jx^{(0)} + h) = x^{(1)}$ and so on iteratively.

To prove the convergence bound, first note that $\frac{\partial}{\partial x_i}\Phi_{MF}(x) = J_i \cdot x + h_i - \tanh^{-1}(x_i)$ and then observe by Lemma 2.3 and concavity that

$$\begin{aligned}
\Phi_{MF}(x^*) - \Phi_{MF}(x^{(t)}) &\le \langle \nabla \Phi_{MF}(x^{(t)}), x^* - x_t \rangle \\
&\le \|\nabla \Phi_{MF}(x^{(t)})\|_1 \\
&= \sum_i |\tanh^{-1}(x_i^{(t)}) - (Jx^{(t)} + h)_i| = \sum_i y_i^{(t)} - y_i^{(t+1)}
\end{aligned}$$

where the second inequality was by Hölder's inequality and $\|x^* - x^{(t)}\|_\infty \le 1$, and the last equality follows from the definition of $y^{(t)}$ and because $y^{(t+1)} \le y^{(t)}$ coordinate-wise. We can also see that $\Phi_{MF}(x^{(t)})$ is a monotonically increasing function of $t$ by considering the path between $x^{(t)}$ and $x^{(t+1)}$ which updates one coordinate at a time, as the gradient always has non-positive entries along this path. Therefore if we sum over $t$ we find that

$$\Phi_{MF}(x^*) - \Phi_{MF}(x^{(T)}) \le \frac{1}{T}\sum_{t=1}^T (\Phi_{MF}(x^*) - \Phi_{MF}(x^{(t)})) \le \frac{1}{T}\sum_{i=1}^n (y_i^{(1)} - y_i^{(T+1)}) \le \frac{\|J\|_1 + \|h\|_1}{T}$$

since $y_i^{(T+1)} \ge 0$ and $y_i^{(1)} \le \sum_j J_{ij} + h_i \le \|J_i\|_1 + h_i$. $\square$

The following simple example shows that the above result is not too far from optimal, in the sense that an asymptotic rate of $o(1/t^2)$ is impossible. We take advantage of the fact that when the model is completely symmetrical, the behavior of the update can be reduced to a 1-dimensional recursion, which is a standard trick (see e.g. [21, 24]).

**Example 2.5.** Consider any $d$-regular graph with no external field and edge weight $\beta = 1/d$, which corresponds to the naive mean field prediction for the critical temperature. By symmetry, analyzing the mean field iteration reduces to the 1d recursion $x \mapsto \tanh(x)$ which behaves like $x \mapsto x - x^3/3$ near the fixed point $x = 0$. Solving this recurrence, we see that $x$ converges to 0 at rate $\Theta(1/\sqrt{t})$. In terms of $x$, the estimated mean field free energy is $(n/2)x^2 + nH(\frac{1+x}{2})$, so by expanding we see that the estimated free energy converges at a $\Theta(1/t^2)$ rate in this example.

## 2.2 A Faster Asymptotic Rate

The above theorem and lower bound leave a gap between $O(1/t)$ and $\Omega(1/t^2)$ for the asymptotic rate of the mean-field iteration. This section is devoted to showing that for large $t$, we can obtain an improved asymptotic rate of $O(1/t^{4/3})$ for the mean-field iteration using a slightly more involved variant of the argument from the previous section. The key insight is that we can obtain some control of $\|x - x^*\|_\infty$ by consider the behavior of higher-order terms when expanding around $x^*$, and this can be used to get better bounds on the convergence in objective.

**Theorem 2.6** (Second bound in Theorem 1.2). *Suppose that $x_0 \in S$ and define $(x_t, y_t)_{t=1}^\infty$ by iterating the mean-field equations. Then for every $t$, $x_t \in S$. Furthermore for any $t \geq 1$,*

$$\|x_t - x^*\|_\infty^3 \leq \frac{\|J\|_1 + \|h\|_1}{t}$$

*and*

$$\Phi_{MF}(x^*) - \Phi_{MF}(x_{2t}) \leq \left(\frac{\|J\|_1 + \|h\|_1}{t}\right)^{4/3}.$$

## 2.3 Aside: Computing the Mean-Field Optimum given Inconsistent Fields

In this section we describe a polynomial time algorithm to compute the optimal mean-field approximation even in the situation when the external fields have inconsistent signs (i.e. some of the $h_i$ are negative, some are positive). This is by reduction to the following algorithmic result of [27], following the same strategy as [16, 33] for the case of the Bethe free energy. We include this result as we were not aware of it appearing explicitly in the literature, though it is known at least as a "folk-lore" result.

**Theorem 2.7.** *Fix an Ising model with ferromagnetic interactions ($J_{ij} \geq 0$) and arbitrary (not necessarily consistent) external field $h$. Then the mean-field free energy $\max_x \Phi_{MF}(x)$ can be approximated within error $\epsilon n$ in time $poly(1/\epsilon, n, \|J\|_1, \|h\|_1)$.*

# 3 Rapid Convergence of Belief Propagation

In this section, we give the proof of our main result Theorem 1.3 by analyzing belief propagation. This proof is considerably more involved than the case of the mean-field iteration; a major conceptual difference between the two iterations is that the mean-field iteration always maintains a valid product distribution, and so can be understood in terms of the landscape of $\Phi_{MF}$, whereas BP operates on "dual" variables which do not correspond to valid pseudodistributions except at fixed points, so analyzing the landscape of $\Phi_{Bethe}$ by itself does not suffice. We get around this by also considering the behavior of a dual functional $\Phi_{Bethe}^*$ which is well-defined for every set of BP messages; however this functional is poorly behaved in general (it can be unbounded and its critical points are typically saddle points). We are able to handle these difficulties by identifying the special behavior of BP and $\Phi_{Bethe}^*$ on two special subsets of the BP messages arising from the partial order structure: the *pre-fixpoints* and *post-fixpoints*. Finally, when analyzing BP in these regions we are able to relate in a useful way its behavior at different values of external field, enabling us to use a convexity argument from [7] which cannot be directly applied to our setting.

Omitted proofs are deferred to Appendices D-F corresponding to each subsection.

## 3.1 Background: BP and the Bethe Free Energy

In this section we recall the necessary facts we need about the relationship derived in [36] between the Bethe free energy and BP. This relationship and corresponding formulas are a bit involved so we sketch the derivation in Appendix D.

Following the correspondence outlined in [36], by writing down the Lagrangian corresponding to the optimization problem (5) over the polytope of locally consistent distributions, one can derive an expression for the Bethe free energy at a critical point of the Lagrangian in terms of the dual variables (Lagrange multipliers) — see [21, 35]. After a change of variables to $\nu$, this lets us define (for all $\nu$, not necessarily fixed points of the BP equations), the *dual Bethe free energy*

$$\Phi^*_{Bethe}(\nu) := \sum_i F_i(\nu) - \sum_{i \sim j} F_{ij}(\nu). \tag{7}$$

where

$$F_i(\nu) = \log \left[ e^{h_i} \prod_{j \in \partial i} (1 + \tanh(J_{ij})\nu_{j \to i}) + e^{-h_i} \prod_{j \in \partial i} (1 - \tanh(J_{ij})\nu_{j \to i}) \right] + \sum_{j \in \partial i} \log \cosh(J_{ij})$$

and $F_{ij}(\nu) = \log(1 + \tanh(J_{ij})\nu_{i \to j}\nu_{j \to i}) + \log \cosh(J_{ij})$. We remark (see [21]) that in the case the graph is a tree, it's known that the Bethe free energy is a convex function, so $\Phi^*_{Bethe}(\nu)$ plus the Lagrange multiplier terms is actually the Lagrangian dual and thus has a natural interpretation for all $\nu$. This is not true in general, however we will soon see that $\Phi^*_{Bethe}$ does have useful properties on some special subsets of the space of messages.

## 3.2 Properties of BP and Optimization Landscape

Throughout the remainder of the paper we will use the notation $\phi(\nu)$ to denote the result of performing a single iteration of belief propagation from $\nu$, i.e.

$$\phi(\nu)_{i \to j} := \tanh \left( h_i + \sum_{k \in \partial i \setminus j} \tanh^{-1}(\tanh(J_{ik})\nu_{k \to i}) \right).$$

The following lemma implies that $\phi(\nu)_{i \to j}$ is a concave monotone function for nonnegative $\nu$.

**Lemma 3.1.** *Suppose that $f(x) = \tanh(h + \sum_i \tanh^{-1}(x_i))$ for any $h \geq 0$. Then $f$ is a concave monotone function on the domain $[0, 1)^n$. Furthermore $\nabla^2 f(x) \prec 0$ unless $h = 0$ and $|\operatorname{supp}(x)| \leq 1$.*

There are two special subsets of the nonnegative messages which will play key roles in our analysis. They are the set of *pre-fixpoints* and *post-fixpoints* (following standard poset terminology), defined by

$$S_{pre} = \{\nu : 0 \leq \phi(\nu)_{i \to j} \leq \nu_{i \to j}\}, \quad S_{post} = \{\nu : 0 \leq \nu_{i \to j} \leq \phi(\nu)_{i \to j}\}.$$

Note that both sets contain the nonnegative fixed points; also we note from Lemma 3.1 that $S_{post}$ is a convex set, whereas $S_{pre}$ is typically non-convex and even disconnected. The gradient of $\Phi^*_{Bethe}$ is well-behaved on these sets:

**Lemma 3.2.** *For any $\nu \geq 0$, $\|\nabla \Phi^*_{Bethe}(\nu)\|_\infty \leq 1$. Furthermore, if $\nu \in S_{pre}$ then $\nabla \Phi^*_{Bethe}(\nu) \leq 0$ and if $\nu \in S_{post}$ then $\nabla \Phi^*_{Bethe}(\nu) \geq 0$.*

The Knaster-Tarski theorem [28] shows that the fixed points of $\phi$ must form a complete lattice, and in particular shows that a greatest fixed point must exist; the following lemma identifies it explicitly.

**Lemma 3.3.** *Suppose that BP is run from initial messages $\nu^{(0)}_{i \to j} = 1$. The messages converge to a fixed point $\nu^*$ of the BP equations such that for any other fixed point $\mu$, $\mu_{i \to j} \leq \nu^*_{i \to j}$ for all $i, j$. Furthermore*

$$\Phi^*_{Bethe}(\nu^*) = \max_{\nu \in S_{post}} \Phi^*_{Bethe}(\nu)$$

The following key theorem states that $\nu^*$ is a global optimum; its proof involves intricate analysis of the Bethe free energy objective and is left to the appendix. The high level idea is similar to the previous situation with mean field approximation — since $J, h$ are nonnegative, based on the form of the Bethe free energy we would guess that the optimizing pseudodistribution exhibits only positive correlations, and that this should be reflected in the optimum BP fixed point having nonnegative messages.

**Theorem 3.4.** *The maximal fixed point $\nu^*$ (as defined in Lemma 3.3) corresponds to a global maximizer of the Bethe free energy.*

### 3.3 Convergence rate of belief propagation

*A priori*, there is no significance to the value of $\Phi^*_{Bethe}$ on a general (non-fixed point) $\nu$. However, we observe that $\Phi^*_{Bethe}$ behaves nicely with respect to BP for messages in $S_{pre}$:

**Lemma 3.5.** *Suppose that $\nu^{(0)} \in S_{pre}$ and define the BP iterates $\nu^{(t+1)} := \phi(\nu^{(t)})$. Then for any $T \geq 0$ and any $\mu$ such that $\nu^{(T)} \leq \mu \leq \nu^{(0)}$ it follows that $\Phi^*_{Bethe}(\mu) \leq \Phi^*_{Bethe}(\nu^{(T)})$. In particular, $\Phi^*_{Bethe}(\nu^{(0)}) \leq \Phi^*_{Bethe}(\nu^{(T)})$.*

In order to give a quantitative bound on the convergence of BP, we are faced with an important conceptual difficulty: the BP messages may not converge quickly in parameter space, but if the BP messages are far from a fixed point in parameter space it is hard to show anything about the quality of their estimate to the free energy. We overcome this difficulty by relating the behavior of BP at zero external field and with additional positive external field, which allows us to take advantage of the smoothness of the primal objective $\Phi_{Bethe}$ — Lemma 3.5 and Theorem 3.4 are key tools needed to make this connection work. This trick is similar in spirit to the use of monotone couplings in the proof of various correlation inequalities for the Ising model. This, combined with a useful concavity argument from [7] for analyzing the positive external field setting, allow us to ultimately prove Theorem 1.3. The detailed proof is in Appendix F.

**Lower bounds and examples:** We give examples which illustrate the importance of distinguishing rates in parameter space vs. objective space, the importance of initialization at all-ones, and also give lower bounds on the asymptotic convergence rate of BP in Appendix G.

**A method with exponentially fast convergence:** As mentioned in the introduction, we use the insights developed in our analysis of BP (especially, the nice structure of the set of the set of *post-fixpoints* $S_{post}$) to give a different method with much faster asymptotic convergence (but with a runtime that depends more significantly on the dimension): this is in Appendix H.

**Acknowledgements:** The author thanks Vishesh Jain for suggesting the argument in Section 2.2, Elchanan Mossel, Nike Sun, Matthew Brennan, and Enric Boix for helpful discussions related to this work, and Ankur Moitra and Andrej Risteski for useful discussions on related topics.

## Footnotes

[1]The equivalence is given by treating the underlying graph as complete with $J_{ij} = 0$ for unconnected nodes, where the optimal coupling of these non-neighbors is when they are independent.

[2]In this theorem and throughout, we use the notation $\|J\|_1, \|J\|_\infty$ to refer to the corresponding $\ell_1, \ell_\infty$ norms of $J$ when viewed as a vector of entries.

[3]For example, if there is no external field then $\mathbb{E}[X_i] = 0$ for all $i$ by symmetry, but e.g. on a 2D lattice at low temperature one can see the optimal BP solution has different marginals [21]. Roughly, this kind of behavior correspond to the existence of *phase transitions* at zero external field (for say the random $d$-regular graph), which are ruled out in the case of strictly positive external field by the Lee-Yang theorem [17].

[4]The convergence in parameter space may be slower due to flat directions of the objective: see Appendix G. This differs from the setting where the external field is lower bounded by a positive constant [7].

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
