[Supplementary Material]

**Supplementary material for "Fast Convergence of Belief Propagation to Global Optima: Beyond Correlation Decay"**

## A    Deferred proofs from Section 2.1

*Proof of Lemma 2.2.* Suppose there exist two critical points $y$ and $z$. Recall that being a critical point is equivalent to solving the mean-field equation $y = \tanh^{\otimes n}(Jy + h)$. Consider the line through $y$ and $z$; this line intersects the boundary region $[0,1]^n \setminus (0,1]^n$ at some point; we parameterize the line as $x(t)$ so that $x(0)$ is on this boundary, i.e. $x(0)_i = 0$ for some $i$, $x(t_1) = y$ and $x(t_2) = z$. Without loss of generality we assume that $t_1 < t_2$. Now we consider the behavior of the function

$$g(t) := \tanh(J_i \cdot x(t) + h_i) - x(t)_i$$

on this line. Observe that by definition $g(0) = \tanh(J_i \cdot x(0) + h_i) - 0 \geq 0$ and $g(t_1) = 0$. It follows from strict concavity that $g(t_2) < 0$ since $t_2 > t_1$, so $z$ cannot be a fixed point, which gives a contradiction. $\qquad\square$

*Proof of Lemma 2.3.* First we claim that $\Phi_{MF}$ is concave at $x^*$. If $x^*$ is on the interior of $[0,1]^n$, then this follows from the second-order optimality condition. From the mean-field equations (first order optimality condition) we see that it's impossible that there are any coordinates such that $x_i^* = 1$, and that if the graph is connected and there is a single coordinate such that $x_i^* = 0$, that the entire vector $x^* = 0$. If $x^* = 0$, then the maximum eigenvalue of $J$ must be 1, so the free energy functional is globally concave – otherwise, by the Perron-Frobenius theorem there exists a eigenvector of $J$ with all nonnegative entries and with eigenvalue greater than 1, from which we see that $x^* = 0$ cannot be the global optimum.

Now, it is easy to see that $\Phi_{MF}$ is concave on all of $S$, becuase if $0 \leq x \leq y$ coordinate-wise then $\nabla^2 \Phi_{MF}(x) \succeq \nabla^2 \Phi_{MF}(y)$, which follows because

$$\nabla^2 \Phi_{MF}(x) - \nabla^2 \Phi_{MF}(y) = (1/4) \sum_i (H''((1+x)/2) - H''((1+y)/2)) e_i e_i^T \succeq 0.$$

since $H''((1+x)/2) = \frac{-2}{1-x^2}$. $\qquad\square$

## B    Deferred proofs from Section 2.2

**Lemma B.1.** *Suppose that $x \in S$. Then*

$$\|\nabla \Phi_{MF}(x)\|_1 \geq \frac{\|x - x^*\|_4^4}{\|x - x^*\|_\infty}$$

*where $x^*$ is as above, the global maximizer of $\Phi_{MF}$ in $[0,1]^n$.*

*Proof.* Recall that

$$\nabla \Phi_{MF}(x) = Jx + h - \sum_i \tanh^{-1}(x_i) e_i.$$

Since $x^*$ is a critical point and local maximum, so $\nabla \Phi_{MF}(x^*) = 0$ and $\nabla^2 \Phi_{MF}(x^*) \preceq 0$, then using that $\frac{d^2}{dx^2} \tanh^{-1}(x) = \frac{2x}{(1-x^2)^2}$, we see that by applying the fundamental theorem of calculus twice that

$$\nabla \Phi_{MF}(x) = J(x - x^*) - \sum_i e_i (\tanh^{-1}(x_i) - \tanh^{-1}(x_i^*)) = \nabla^2 \Phi_{MF}(x^*)(x - x^*) - \sum_i e_i \int_{x_i^*}^{x_i} \int_{x_i^*}^z \frac{2y}{(1-y^2)^2} dy dz$$

and so

$$\langle x^* - x, \nabla \Phi_{MF}(x) \rangle \geq \sum_i (x_i - x_i^*) \int_{x_i^*}^{x_i} \int_{x_i^*}^z \frac{2y}{(1-y^2)^2} dy dz$$

$$\geq \sum_i (x_i - x_i^*) \int_{x_i^*}^{x_i} \int_{x_i^*}^z 2y \, dy dz$$

$$= \sum_i (x_i - x_i^*)(x_i^3/3 - (x_i^*)^3/3 - (x_i - x_i^*)(x_i^*)^2)$$

$$= \sum_i (x_i - x_i^*)^2 (x_i^2 + x_i x_i^*) \geq \sum_i (x_i - x_i^*)^4$$

where in the last inequality we used $x_i \geq x_i^* \geq 0$. Finally the result follows combining the above with $\langle x^* - x, \nabla \Phi_{MF}(x) \rangle \leq \|x^* - x\|_\infty \|\nabla \Phi_{MF}\|_1$ by Hölder's inequality. $\square$

*Proof of Theorem 2.6.* From Lemma B.1 we see that

$$\|x - x^*\|_\infty^3 \leq \frac{\|x - x^*\|_4^4}{\|x - x^*\|_\infty} \leq \|\nabla \Phi_{MF}(x)\|_1$$

and so as in the proof of Theorem 2.4 we see that for any $T$,

$$\|x_T - x^*\|_\infty^3 \leq \frac{1}{T}\sum_{t=1}^T \|x_t - x^*\|_\infty^3 \leq \frac{1}{T}\sum_{t=1}^T \|\nabla \Phi_{MF}(x_t)\|_1 = \frac{1}{T}\sum_{i=1}^n (y_{1,i} - y_{T+1,i}) = \frac{\|J\|_1 + \|h\|_1}{T}.$$

Therefore for any $t' > T$ we see by convexity and Hölder's inequality

$$\Phi_{MF}(x^*) - \Phi_{MF}(x_t) \leq \langle \nabla \Phi_{MF}(x_t), x^* - x_t \rangle \leq \left( \frac{\|J\|_1 + \|h\|_1}{T} \right)^{1/3} \|\nabla \Phi_{MF}(x_t)\|_1$$

$$= \left( \frac{\|J\|_1 + \|h\|_1}{T} \right)^{1/3} \sum_i |\tanh^{-1}(x_{t,i}) - (Jx_t + h)_i|$$

$$= \left( \frac{\|J\|_1 + \|h\|_1}{T} \right)^{1/3} \sum_i (y_{t,i} - y_{t+1,i})$$

and summing this over $t' = T + 1$ to $2T$ and telescoping we see that

$$\Phi_{MF}(x^*) - \Phi_{MF}(x_{2T}) \leq \frac{1}{T}\sum_{t'=T+1}^{2T} (\Phi_{MF}(x^*) - \Phi_{MF}(x_{t'})) \leq \left( \frac{\|J\|_1 + \|h\|_1}{T} \right)^{1/3} \sum_i (y_{T,i} - y_{2T,i})$$

$$\leq \left( \frac{\|J\|_1 + \|h\|_1}{T} \right)^{4/3}$$

which proves the result. $\square$

## C   Deferred proofs from Section 2.3

**Theorem C.1** ([27]). *Let $(\Sigma, \leq)$ be a finite alphabet equipped with a total ordering, fix a finite graph $G = (V, E)$ and fix functions $f_v : \Sigma \to \mathbb{R}$ and $f_{u,v} : \Sigma \times \Sigma \to \mathbb{R}$. Suppose that every $f = f_{u,v}$ satisfies the following submodularity condition:*

$$f(\min(x_1, x_2), \min(y_1, y_2)) + f(\max(x_1, x_2), \max(y_1, y_2)) \leq f(x_1, y_1) + f(x_2, y_2)$$

*Then the optimization problem*

$$\min_{L:V \to \Sigma} \left[ \sum_{v \in V} f_v(L(v)) + \sum_{u \sim v} f_{u,v}(L(u), L(v)) \right].$$

*is efficiently solvable in time $poly(|\Sigma|, |V|)$.*

*Proof of Theorem 2.7.* Supermodularity (which will become submodularity after converting to a minimization problem by negating) for the edge interactions $J_{ij} x_i x_j$ is immediate, so to apply this theorem all we need to do is discretize the optimization problem $\max \Phi_{MF}$ appropriately: to do this we compute Lipschitz constants on the relevant part of the space. First observe that

$$|x_i^*| = |\tanh(J_i \cdot x_{\sim i}^* + h_i)| \leq \tanh(\|J_i\|_1 + |h_i|)$$

so if we restrict $x_i$ to lie within $[-\tanh(\|J_i\|_1 + |h_i|), \tanh(\|J_i\|_1 + |h_i|)]$ then

$$\max_{x_i} |\frac{d}{dx_i} H(Ber(\frac{1+x_i}{2}))| = \max_{x_i} \tanh^{-1}(\tanh(\|J_i\|_1 + |h_i|)) = |J_i|_1 + |h_i|$$

so $H(x_i)$ is $(\|J_i\|_1 + |h_i|)$-Lipschitz on this interval. Similarly we observe that

$$\left| \sum_j J_{ij}x_j + h_i \right| \leq \|J_i\|_1 + |h_i|$$

so if we discretize each coordinate with grid size $\frac{\epsilon}{2(\|J_i\|_1 + |h_i|)}$ then we change the objective by at most $\epsilon n$. Then by the result of [27] this problem can be solved to optimality in time $poly(1/\epsilon, n, \max_i \|J_i\| + |h_i|)$. $\qquad\square$

# D   Deferred material from Section 3.1

In this section we quickly recall the basic definitions, facts, and notations involving the Bethe free energy, its dual formulation, and the connection to Belief Propagation – as described in [36] and reference text [21]. We repeat some of the standard calculations in order to express the results in our variables $\nu_{i\to j}$. The Lagrangian corresponding to the optimization problem (5) over the polytope of locally consistent distributions (which is defined over all, not necessarily consistent, $P_{ij}$ and $P_i$) is

$$\mathcal{L}(P,\lambda) = \Phi_{Bethe}(P) + \sum_{i,j,x_i} \lambda_{i\to j}(x_i)(\sum_{x_j} P_{ij}(x_i,x_j) - P_i(x_i)) + \sum_i \lambda_i(\sum_{x_i} P_i(x_i) - 1)$$

where we ignore the constraint $P_i(x_i) \geq 0$ because, given the other constraints, this constraint is always satisfied at a critical point (since the derivative of $H(Ber(p))$ diverges as $p \to 0$ or $p \to 1$).

By differentiating with respect to $P$, and setting $\lambda'_{i\to j} = \frac{\lambda_{i\to j}(1) - \lambda_{i\to j}(-1)}{2}$, one finds that at a critical point of the Lagrangian that

$$P_{ij}(x_i,x_j) \propto e^{J_{ij}x_ix_j + \lambda_{i\to j}(x_i) + \lambda_{j\to i}(x_j)} \propto e^{J_{ij}x_ix_j + \lambda'_{i\to j}x_i + \lambda'_{j\to i}x_j}$$

and

$$P_i(x_i) \propto \exp\left(\frac{1}{\deg(i)-1}\sum_j \lambda_{i\to j}(x_i) - \frac{h_i}{\deg(i)-1}x_i\right) \propto \exp\left(\frac{1}{\deg(i)-1}\sum_j \lambda'_{i\to j}x_i - \frac{h_i}{\deg(i)-1}x_i\right).$$

Furthermore by differentiating with respect to $\lambda$ we see that the constraints are satisfied, therefore for any $i \sim j$ that $P_i(x_i) = \sum_{x_j} P_{ij}(x_i,x_j)$ hence

$$P_i(x_i)^{\deg(i)-1} \propto \prod_{k\in\partial i\setminus\{j\}} \sum_{x_k} P_{ik}(x_i,x_k) \propto \sum_{x_{\partial i\setminus j}} e^{\sum_k(J_{ik}x_ix_k + \lambda'_{i\to k}x_i + \lambda'_{k\to i}x_k)} = e^{\sum_k \lambda'_{i\to k}x_i} \sum_{x_{\partial i\setminus j}} e^{\sum_k(J_{ik}x_i + \lambda'_{k\to i})x_k}$$

so

$$e^{\lambda'_{i\to j}x_i - h_ix_i} \propto \sum_{x_{\partial i\setminus j}} e^{\sum_k(J_{ik}x_i + \lambda'_{k\to i})x_k} \propto \prod_k \sum_{x_k} e^{J_{ik}x_i} e^{\lambda'_{k\to i}x_k}.$$

Define $\nu_{i\to j} := \tanh(\lambda'_{i\to j})$ so $\frac{1 + \nu_{i\to j}x_i}{2} = \frac{e^{\lambda'_{i\to j}x_i}}{e^{\lambda'_{i\to j}} + e^{-\lambda'_{i\to j}}}$, then we see

$$\begin{aligned}
\nu_{i\to j} &= \frac{e^{h_i}\prod_k \sum_{x_k} e^{J_{ik}x_k}e^{\lambda'_{k\to i}x_k} - e^{-h_i}\prod_k \sum_{x_k} e^{-J_{ik}x_k}e^{\lambda'_{k\to i}x_k}}{e^{h_i}\prod_k \sum_{x_k} e^{J_{ik}x_k}e^{\lambda'_{k\to i}x_k} + e^{-h_i}\prod_k \sum_{x_k} e^{-J_{ik}x_k}e^{\lambda'_{k\to i}x_k}} \\
&= \frac{e^{h_i}\prod_k \sum_{x_k} e^{J_{ik}x_k}(1 + \nu_{k\to i}x_k) - e^{-h_i}\prod_k \sum_{x_k} e^{-J_{ik}x_k}(1 - \nu_{k\to i}x_k)}{e^{h_i}\prod_k \sum_{x_k} e^{J_{ik}x_k}(1 + \nu_{k\to i}x_k) + e^{-h_i}\prod_k \sum_{x_k} e^{-J_{ik}x_k}(1 - \nu_{k\to i}x_k)} \\
&= \tanh(h_i + \sum_{k\in\partial i\setminus j} \tanh^{-1}(\tanh(J_{ik})\nu_{k\to i}))
\end{aligned}$$

which is the form of the BP equation we will typically refer to. We will denote the right hand side by $\phi(\nu)_{i\to j}$ so the BP iteration is given by $\nu \mapsto \phi(\nu)$. We will also define $\theta_{ik} = \tanh(J_{ik})$. Finally,

we will explicitly rewrite the Bethe free energy at a critical point in terms of the messages $\nu_{i \to j}$. We claim that at a critical point $(P, \lambda)$

$$\Phi_{Bethe} = \sum_i F_i(\lambda) - \sum_{i \sim j} F_{ij}(\lambda)$$

where

$$F_i(\lambda) := \log \sum_{x_i} e^{h_i x_i} \prod_{j \in \partial i} \sum_{x_j} e^{J_{ij} x_i x_j} e^{\lambda'_{j \to i} x_j} = \log \sum_{x_i, x_{\partial i}} e^{h_i x_i + \sum_j J_{ij} x_i x_j + \lambda'_{j \to i} x_j}$$

$$F_{ij}(\lambda) := \log \sum_{x_i, x_j} e^{J_{ij} x_i x_j + \lambda'_{i \to j} x_i + \lambda'_{j \to i} x_j}.$$

To see this observe that from the Gibbs variational principle that (considering a joint distribution where we sample $X_i$ from $P_i$ and then $X_j | X_i$ according to $P_{ij}$)

$$F_i(\lambda) = \mathbb{E}[h_i X_i + \sum_j J_{ij} X_i X_j + \lambda'_{j \to i} X_j] + H(X_i) + \sum_{j \in \partial i} H(X_j | X_i)$$

$$= \mathbb{E}[h_i X_i + \sum_j J_{ij} X_i X_j + \lambda'_{j \to i} X_j] + \sum_{j \in \partial i} H(X_i, X_j) - (\deg(i) - 1)H(X_i)$$

and

$$F_{ij}(\lambda) = \mathbb{E}[\sum_j J_{ij} X_i X_j + \lambda'_{i \to j} X_i + \lambda'_{j \to i} X_j] + H(X_i, X_j)$$

so summing all of these terms does indeed give $\Phi_{Bethe}(P)$. Finally we rewrite in terms of $\nu_{i \to j}$ to get

$$F_i(\nu) = \log \left[ e^{h_i} \prod_{j \in \partial i} \sum_{x_j} e^{J_{ij} x_j} \frac{1 + \nu_{j \to i} x_j}{2} + e^{-h_i} \prod_{j \in \partial i} \sum_{x_j} e^{-J_{ij} x_j} \frac{1 + \nu_{j \to i} x_j}{2} \right]$$

$$= \log \left[ e^{h_i} \prod_{j \in \partial i} (1 + \tanh(J_{ij}) \nu_{j \to i}) + e^{-h_i} \prod_{j \in \partial i} (1 - \tanh(J_{ij}) \nu_{j \to i}) \right] + \sum_{j \in \partial i} \log \frac{e^{J_{ij}} + e^{-J_{ij}}}{2}$$

and

$$F_{ij}(\nu) = \log \sum_{x_i} \sum_{x_j} e^{J_{ij} x_i x_j} \frac{1 + \nu_{i \to j} x_i}{2} \frac{1 + \nu_{j \to i} x_j}{2}$$

$$= \log \left( \frac{e^{J_{ij}} + e^{-J_{ij}}}{2} + \frac{(e^{J_{ij}} - e^{-J_{ij}}) \nu_{i \to j} \nu_{j \to i}}{2} \right)$$

$$= \log(1 + \tanh(J_{ij}) \nu_{i \to j} \nu_{j \to i}) + \log \left( \frac{e^{J_{ij}} + e^{-J_{ij}}}{2} \right).$$

# E   Deferred proofs from Section 3.2

*Proof of Lemma 3.1.* Observe that

$$\frac{\partial f}{\partial x_i}(x) = \frac{1 - f(x)^2}{1 - x_i^2} \geq 0$$

which proves monotonicity, and

$$\frac{\partial^2 f}{\partial x_i \partial x_j}(x) = \frac{-2f(x)(1 - f(x)^2)}{(1 - x_i^2)(1 - x_j^2)} + \mathbb{1}(i = j)\frac{(1 - f(x)^2)2x_i}{(1 - x_i^2)^2} = \frac{(2x_i \mathbb{1}(i = j) - 2f(x))(1 - f(x)^2)}{(1 - x_i^2)(1 - x_j^2)}.$$

Note that for any vector $w$, if $w_i' = (1 - f(x)^2)w_i/(1 - x_i^2)$ then

$$\sum_{ij} w_i \frac{\partial^2 f}{\partial x_i \partial x_j}(x) w_j = \sum_{ij} w_i'(2x_i \mathbb{1}(i = j) - 2f(x))w_j' = \sum_i 2x_i(w_i')^2 - 2f(x)(\sum_i w_i')^2 \leq 0$$

since $x_i \leq f(x)$, which proves concavity. If $h > 0$ or $|\operatorname{supp}(x)| \geq 2$ then this is a strictly inequality since $x_i < f(x)$. $\qquad\square$

*Proof of Lemma 3.2.* This will follow once we compute the gradient of $\Phi^*_{Bethe}(\nu)$. Recall that we defined $\theta_{ij} = \tanh(J_{ij})$. Observe that

$$
\begin{aligned}
\frac{\partial \Phi^*_{Bethe}}{\partial \nu_{i \to j}}(\nu) &= \frac{\partial F_j}{\partial \nu_{i \to j}} - \frac{\partial F_{ij}}{\partial \nu_{i \to j}} \\
&= \frac{e^{h_j}\theta_{ij}\prod_{k \in \partial j \setminus i}(1 + \theta_{jk}\nu_{k \to j}) - e^{-h_j}\theta_{ij}\prod_{k \in \partial j \setminus i}(1 - \theta_{jk}\nu_{k \to j})}{e^{h_j}\prod_{k \in \partial j}(1 + \theta_{jk}\nu_{k \to j}) + e^{-h_j}\prod_{k \in \partial j}(1 - \theta_{jk}\nu_{k \to j})} - \frac{\theta_{ij}\nu_{j \to i}}{1 + \theta_{ij}\nu_{i \to j}\nu_{j \to i}} \\
&= \frac{1}{\nu_{i \to j} + 1/(\theta_{ij}\phi(\nu)_{j \to i})} - \frac{1}{\nu_{i \to j} + 1/(\theta_{ij}\nu_{j \to i})}
\end{aligned}
\tag{8}
$$

where (as defined earlier) $\phi(\nu)_{j \to i}$ denotes the next BP message from $j$ to $i$ based on the current $\nu$. As long as $\nu \geq 0$, we see that

$$
\begin{aligned}
\left| \frac{1}{\nu_{i \to j} + 1/(\theta_{ij}\phi(\nu)_{j \to i})} - \frac{1}{\nu_{i \to j} + 1/(\theta_{ij}\nu_{j \to i})} \right| &= \left| \int_{1/(\theta_{ij}\phi(\nu)_{j \to i})}^{1/(\theta_{ij}\nu_{j \to i})} \frac{1}{(\nu_{i \to j} + x)^2} dx \right| \\
&\leq \left| \int_{1/(\theta_{ij}\phi(\nu)_{j \to i})}^{1/(\theta_{ij}\nu_{j \to i})} \frac{1}{x^2} dx \right| = \theta_{ij}|\nu_{j \to i} - \phi(\nu)_{j \to i}| \leq 1
\end{aligned}
$$

which proves that $\|\nabla\Phi^*_{Bethe}(\nu)\|_\infty \leq 1$. If $\nu \in S_{pre}$ or $S_{post}$ then the signs are determined by (8) as claimed. $\qquad\square$

*Proof of Lemma 3.3.* If $\nu^{0)}_{i \to j}$ and $\nu^{(t)} := \phi(\nu^{(t-1)}$ then from monotonicity of $\phi$ (see Lemma 3.1) we see this is a coordinate-wise decreasing sequence, which must converge to some fixed point. By monotonicity and induction we also see that for any fixed point $\mu$, $\mu_{i \to j} \leq \nu^{(t)}_{i \to j}$ for all $t$, hence for $\nu^*$ as well. Finally, consider any other point $\nu \in S_{post}$: by convexity of $S_{post}$ we see that the line segment from $\nu$ to $\nu^*$ is entirely contained in $S_{post}$, by Lemma 3.2 we see that for any $x$ on this interpolating line that $\nabla\Phi^*_{Bethe}(x) \cdot (\nu^* - \nu) \geq 0$, and integrating this gives that $\Phi^*_{Bethe}(\nu) \leq \Phi^*_{Bethe}(\nu^*)$ as desired. $\qquad\square$

As our final preparation for the theorem, we establish that at least one optimal BP fixed point has only nonnegative messages. First we need the following technical lemma, which allows us to reason about the behavior of the optimal couplings in the Bethe approximation. The realization that solving for this coupling analytically is feasible is due originally to [34], although we parameterize the solution differently.

**Lemma E.1.** *Suppose that $\mathbb{E}[X] \geq 0$ and $\mathbb{E}[Y] \geq 0$ where $X, Y$ are valued in $\{\pm 1\}$. Then*

$$
\max_{coupling} [-\beta \mathrm{Cov}(X, Y) + H(X, Y)] \leq \max_{coupling} [\beta \mathrm{Cov}(X, Y) + H(X, Y)]
$$

*where the maximum ranges over all couplings (i.e. possible joint distributions) $P$ of $X$ and $Y$.*

*Proof.* By subtracting $H(Y)$ on both sides we reduce to showing

$$
\max_{coupling} [-\beta \mathrm{Cov}(X, Y) + H(X|Y)] \leq \max_{coupling} [\beta \mathrm{Cov}(X, Y) + H(X|Y)].
\tag{9}
$$

We will do this by showing both sides are differentiable w.r.t. $\beta$ and that the derivative of the rhs ($\mathrm{Cov}(X, Y)$ at the optimal coupling for the rhs) is larger than the derivative of the lhs ($-\mathrm{Cov}(X, Y)$ for the optimal coupling for the lhs), so that integrating gives the desired inequality.

First we characterize the optimizer of the rhs of (9). Let $\rho := \mathrm{Cov}(X, \frac{Y}{\mathrm{Var}(Y)})$. Then $\mathbb{E}[X|Y] = \mathbb{E}[X] + \rho(Y - \mathbb{E}Y)$ since $\mathrm{Cov}(\mathbb{E}[X|Y], Y) = \mathrm{Cov}(X, Y) = \rho$. Thus the objective maximized by the rhs is

$$
f(\rho) := \beta\mathrm{Var}(Y)\rho + \mathbb{E}_Y H(\mathbb{E}[X] + \rho(Y - \mathbb{E}Y)).
$$

where $H(x) := H(Ber(\frac{1+x}{2}))$. Differentiating in $\rho$ we see that the optimum is when

$$
\mathrm{Var}(Y)\beta = \mathbb{E}_Y[(Y - \mathbb{E}Y)\tanh^{-1}(\mathbb{E}[X] + \rho(Y - \mathbb{E}Y))] = \mathrm{Cov}(Y, \tanh^{-1}(\mathbb{E}[X] + \rho(Y - \mathbb{E}Y))).
$$

Let this relation define $\rho(\beta) \geq 0$. Similarly, define $\rho'(\beta) \geq 0$ to be the solution to $-\text{Var}(Y)\beta = \text{Cov}(Y, \tanh^{-1}(\mathbb{E}[X] - \rho'(Y - \mathbb{E}Y)))$ i.e.

$$\text{Var}(Y)\beta = \text{Cov}(Y, \tanh^{-1}(-\mathbb{E}[X] + \rho'(Y - \mathbb{E}Y)))$$

We now claim that $\rho(\beta) \geq \rho'(\beta)$. As previously described, if we show this then by integrating w.r.t. $\beta$ we get the final inequality. To prove the claim, first subtract the above terms to get that

$$0 = \text{Cov}[Y, \tanh^{-1}(\mathbb{E}[X] + \rho(Y - \mathbb{E}Y)) - \tanh^{-1}(-\mathbb{E}[X] + \rho'(Y - \mathbb{E}Y))].$$

Since the covariance is 0 and $Y$ takes on only two values, it means that the rhs is independent of $Y$, therefore

$$\tanh^{-1}(\mathbb{E}[X]+\rho(1-\mathbb{E}Y))-\tanh^{-1}(-\mathbb{E}[X]+\rho'(1-\mathbb{E}Y)) = \tanh^{-1}(\mathbb{E}[X]+\rho(-1-\mathbb{E}Y))-\tanh^{-1}(-\mathbb{E}[X]+\rho'(-1-\mathbb{E}Y))$$

and rearranging we get

$$\tanh^{-1}(\mathbb{E}[X]+\rho(1-\mathbb{E}Y))-\tanh^{-1}(\mathbb{E}[X]+\rho(-1-\mathbb{E}Y)) = \tanh^{-1}(-\mathbb{E}[X]+\rho'(1-\mathbb{E}Y))-\tanh^{-1}(-\mathbb{E}[X]+\rho'(-1-\mathbb{E}Y))$$

Define $g(x, r) := \tanh^{-1}(x + r) - \tanh^{-1}(x - r)$ for $x, r$ s.t. $|x| + |r| < 1$ and $r \geq 0$. Then the above equation says $g(\mathbb{E}[X] - \rho\mathbb{E}[Y], \rho) = g(-\mathbb{E}[X] - \rho'\mathbb{E}[Y], \rho')$. We obsere that $g$ is even, strictly increasing in $r$, and strictly increasing in $x$ for $x \geq 0$ since $\frac{\partial}{\partial x} g(x, r) = \frac{1}{1-(x+r)^2} - \frac{1}{1-(x-r)^2} \geq 0$. Since $g$ is an even function, we have

$$g(|\mathbb{E}[X] - \rho\mathbb{E}[Y]|, \rho) = g(|\mathbb{E}[X] + \rho'\mathbb{E}[Y]|, \rho'). \tag{10}$$

Suppose $\rho < \rho'$, then because $g$ is a strictly increasing function in both $x \geq 0$ and $r$ we see that the lhs of (10) is strictly less than the rhs, which is a contradiction. Therefore $\rho \geq \rho'$. $\qquad \square$

**Lemma E.2.** *There exists a BP fixed point in $[0, 1)^n$ which corresponds to a global maximizer of the Bethe free energy.*

*Proof.* Observe that for a locally consistent distribution $P$, the Bethe free energy can be rewritten to give

$$\Phi_{Bethe}(P) = \frac{1}{2}\mathbb{E}[X]^T J\mathbb{E}[X] + \sum_i h_i \mathbb{E}[X_i] + \sum_i H(X_i) + \sum_{i \sim j}(J_{ij}\text{Cov}(X_i, X_j) + H(X_i, X_j) - H(X_i) - H(X_j)).$$

We first claim that there exists a global maximizer of this functional (over all locally consistent distributions) satisfying $\mathbb{E}[X_i] \geq 0$ for all $i$. To see this, we consider a fixed feasible local distribution $P$ and claim that there exists $Q$ with sign-flipped marginals $\mathbb{E}_Q[X_i] = |\mathbb{E}_P[X_i]|$ and no smaller value of $J_{ij}\text{Cov}(X_i, X_j) + H(X_i, X_j)$. We now describe the couplings along each edge $i \sim j$: if neither or both of $i$ and $j$ were sign-flipped, then we can simply use the same/sign-flipped coupling from before. Now suppose (w.l.o.g.) that $j$ has the same marginal and $i$ was sign-flipped. Then it follows immediately from Lemma E.1 that there exists a coupling $Q_{ij}$ between $X_i$ and $X_j$ s.t. $J_{ij}\text{Cov}_{Q_{ij}}(X_i, X_j) + H_{Q_{ij}}(X_i, X_j) \geq J_{ij}\text{Cov}_P(X_i, X_j) + H_P(X_i, X_j)$.

Recall that at a critical point of the Lagrangian, for any edge $i \sim j$

$$P_{ij}(x_i, x_j) \propto e^{J_{ij}x_i x_j + \lambda'_{i \to j}x_i + \lambda'_{j \to i}x_j}.$$

Since we have shown that there exists a global maximizer $P$ such that $\mathbb{E}[X_i], \mathbb{E}[X_j] \geq 0$ it must be that at least one of $\lambda'_{i \to j}, \lambda'_{j \to i} \geq 0$. We now show that there exists another locally consistent distribution $Q$ with $\Phi_{Bethe}(Q) \geq \Phi_{Bethe}(P)$ and with corresponding $\lambda'_{i \to j}, \lambda'_{j \to i} \geq 0$ for all edges $i$ and $j$.

The construction of $Q$ goes through the dual free energy $\Phi^*_{Bethe}$. First recall that $\Phi_{Bethe}(P) = \Phi^*_{Bethe}(\nu)$ where $\nu_{i \to j} = \tanh(\lambda'_{i \to j})$ is a fixed point of the BP equations. Furthermore, recall from (8) that

$$\partial_{\nu_{i \to j}}\Phi^*_{Bethe}(\nu) = \frac{1}{\nu_{i \to j} + 1/(\theta_{ij}\phi(\nu)_{j \to i})} - \frac{1}{\nu_{i \to j} + 1/(\theta_{ij}\nu_{j \to i})}.$$

Based on this we claim that for $\mu_{i \to j} = |\nu_{i \to j}|$, $\Phi^*_{Bethe}(\mu) \geq \Phi^*_{Bethe}(\nu)$. We consider flipping one negative coordinate $\nu_{i \to j}$ to $|\nu_{i \to j}|$ at a time and show $\Phi^*_{Bethe}$ is non-decreasing under this operation. First we compute the change using (8):

$$\Phi^*_{Bethe}(\nu_{\sim(i \to j)}, |\nu_{i \to j}|) - \Phi^*_{Bethe}(\nu) = \int_{-|\nu_{i \to j}|}^{|\nu_{i \to j}|} \frac{\partial \Phi^*_{Bethe}}{\partial \nu_{i \to j}} d\nu_{i \to j}$$

$$= \log \frac{|\nu_{i \to j}| + 1/(\theta_{ij}\phi(\nu)_{j \to i})}{-|\nu_{i \to j}| + 1/(\theta_{ij}\phi(\nu)_{j \to i})} - \log \frac{|\nu_{i \to j}| + 1/(\theta_{ij}\nu_{j \to i})}{-|\nu_{i \to j}| + 1/(\theta_{ij}\nu_{j \to i})}$$

$$= \log \frac{1 + |\nu_{i \to j}|\theta_{ij}\phi(\nu)_{j \to i}}{1 - |\nu_{i \to j}|\theta_{ij}\phi(\nu)_{j \to i}} - \log \frac{1 + |\nu_{i \to j}|\theta_{ij}\nu_{j \to i}}{1 - |\nu_{i \to j}|\theta_{ij}\nu_{j \to i}}.$$

Finally, we notice that this expression is nonnegative as long as $\phi(\nu)_{j \to i} \geq \nu_{j \to i} \geq 0$. Recall that if we are flipping $\nu_{i \to j}$ from negative to positive, by our previous argument it is guaranteed that $\nu_{j \to i} \geq 0$. Furthermore, initially we start from a BP fixed point so $\phi(\nu)_{j \to i} = \nu_{j \to i}$, and increasing coordinates of $\nu$ only increases $\phi(\nu)$, so we maintain the invariant $\phi(\nu)_{j \to i} \geq \nu_{j \to i}$ for all $j, i$ except possibly for those $\nu_{j \to i}$ which have been previously flipped, in which case there is no issue because we will never flip $\nu_{i \to j}$.

Therefore $\mu$ indeed satisfies that $\Phi^*_{Bethe}(\mu) \geq \Phi^*_{Bethe}(\nu)$, and also from the definition we see that $\mu'_{i \to j} \geq |\phi(\nu)_{i \to j}| = |\nu_{i \to j}| = \mu_{i \to j}$ so $\mu \in S_{post}$. Therefore by Lemma 3.3 we see $\Phi^*_{Bethe}(\mu^*) \geq \Phi^*_{Bethe}(\mu) \geq \Phi^*_{Bethe}(\nu)$. Hence $\mu^*$ is a BP fixed point which corresponds to a locally consistent distribution $Q$ with $\Phi_{Bethe}(Q) = \Phi^*_{Bethe}(\mu^*) \geq \Phi^*_{Bethe}(\nu) = \Phi_{Bethe}(P)$, so $Q$ is a global maximizer of $\Phi_{Bethe}$, and $\mu^*_{i \to j} \geq 0$ for all $i$ and $j$.

$\square$

*Proof of Theorem 3.4.* By Lemma E.2 there exists some $\mu$ with $\mu_{i \to j} \geq 0$ for all $i, j$ such that $\Phi^*_{Bethe}(\mu)$ equals the global maximum of the Bethe free energy. However, by Lemma 3.3, the fixed point $\nu^*$ satisfies $\Phi^*_{Bethe}(\nu^*) \geq \Phi^*_{Bethe}(\mu)$. Therefore the locally consistent distribution $P$ corresponding to $\nu^*$ (which satisfies $\Phi^*_{Bethe}(\nu^*) = \Phi_{Bethe}(\nu)$) must be a global maximizer of the Bethe free energy. $\square$

# F  Deferred proofs from Section 3.3

*Proof of Lemma 3.5.* We prove this by constructing a coordinate-wise monotonically decreasing path from $\mu$ to $\nu^{(T)}$ contained in $S_{pre}$. Then, because the gradient is coordinate-wise nonpositive in $S_{pre}$ due to Lemma 3.2, it follows that the first derivative of $\Phi^*$ along (each segment of) this path is nonnegative which proves the inequality by integration.

We construct this path segment-by-segment using an iterative process. For any $\nu$ such that $\nu^{(T)} \leq \nu \leq \nu^{(0)}$, define $T(\nu, i, j) = T_{\nu_0}(\nu, i, j) := \max\{t \geq 0 : \nu_{i \to j} \leq \nu_{i \to j}^{(t)}\}$. In other words, $T_{\nu_0}(\nu, i, j)$ is the last time at which BP iterated from $\nu^{(0)}$ has a larger message from $i \to j$ than in the specified $\nu$.

1. Let $\mu(0) = \mu$ and set $s := 0$.

2. While there exists $i$ and $j$ such that $T(\mu(s), i, j) < T$:

   (a) Choose $i$ and $j$ which minimize $t := T(\mu(s), i, j)$.

   (b) Define $\mu(s+1)_{i \to j} = \nu_{i \to j}^{(t+1)}$ and $\mu(s+1) = \mu(s)$ in all other coordinates. For $s' \in (s, s+1)$ define $\mu(s')$ by linearly interpolating $\mu(s)$ and $\mu(s+1)$.

   (c) Set $s := s + 1$.

Note that this process maintains the invariant $\mu(s) \geq \nu^{(t)}$ and that at each step of the above process, we increase $T(\mu(s), i, j)$ by 1 so the process must terminate in a finite number of steps with the path $\mu(\cdot)$ terminating at $\nu^{(T)}$. It remains to check that this process stays inside of $S_{pre}$ which we check by induction. Given that $\mu(s) \in S_{pre}$, let $t$ be as defined in step 2 (a) above and let $\mu'$ be any linear interpolate between $\mu(s)$ and $\mu(s+1)$. Then we know $\mu(s) \leq \nu^{(t)}$ so $\phi(\mu(s)) \leq \phi(\nu^{(t)}) = \nu^{(t+1)}$

hence $\phi(\mu')_{i\to j} \leq \phi(\mu(s))_{i\to j} \leq \nu_{i\to j}^{(t+1)} = \mu(s+1)_{i\to j} \leq \mu'_{i\to j}$. For the other coordinates $a, b$ it's immediate from monotonicity and the induction hypothesis that $\phi(\mu')_{a\to b} \leq \phi(\mu(s))_{a\to b} \leq \mu(s)_{a\to b} = \mu'_{a\to b}$ so $\mu' \in S_{pre}$. $\qquad\square$

The following Lemma gives the bound (in parameter space) for BP at positive external field which we will use; it is a variant of Lemma 4.3 from [7] which is more optimized for our use. It is convenient to rephrase the result of Ising models on trees, in which case it gives a quantitative bound showing that under positive external field, the root marginal on an infinite tree does not distinguish between all-plus and free boundary conditions. The connection to our problem is that, because BP computes exact marginals on trees, Loopy BP computes true marginals on its corresponding "computation tree" which is a truncation of the non-backtracking walk tree, with boundary conditions on the bottom level determined by its initialization.

**Lemma F.1** (Variant of Lemma 4.3 from [7]). *Suppose $T$ is an infinite tree rooted at $\rho$ and suppose the minimum external field is $h_{min} := \min_i h_i > 0$. Then*

$$\mathbb{E}_{T(\ell)}[X_\rho | X_{T(\ell)} = 1] - \mathbb{E}_{T(\ell)}[X_\rho] \leq \frac{1 + \|J\|_\infty)}{\ell \tanh(h_{min})}$$

*where $\mathbb{E}_{T(\ell)}$ denotes the expectation under the measure where the tree is truncated at level $\ell$.*

*Proof.* The proof is largely the same as in [7] with a slight difference in the bounds. Observe that $\mathbb{E}_{T(\ell)}[X_\rho | X_{T(\ell)} = 1]$ is the same as $\mathbb{E}_{T(\ell-1),B}[X_\rho]$ where $B$ corresponds to additional external field $\sum_{j \in C(i)} J_{ij}$ at every node $i$ on level $\ell - 1$. Similarly, $\mathbb{E}_{T(\ell)}[X_\rho]$ is the same as $\mathbb{E}_{T(\ell-1),h}[X_\rho]$ where there is additional field $B'_i$ of $\sum_{j \in C(i)} \tanh^{-1}(\tanh(J_{ij})\tanh(h_j))$ at the leaves of $T(\ell-1)$. Define

$$M := \sup_{i \in T, j \in C(i)} \frac{J_{ij}}{\tanh^{-1}(\tanh(J_{ij})\tanh(h_j))} \leq \sup_{i \in T, j \in C(i)} \frac{J_{ij}}{\tanh(J_{ij})\tanh(h_{min})} \leq \frac{1 + \|J\|_\infty}{\tanh(h_{min})}$$

where we used the inequalities $\tanh^{-1}(x) \geq x$ and $x/\tanh(x) \leq 1 + x$ for $x \geq 0$.

Note from above that $\mathbb{E}_{T(\ell)}[X_\rho] = \mathbb{E}_{T(\ell-1),h}[X_\rho] \geq \mathbb{E}_{T(\ell-1)}[X_\rho]$ by Griffith's inequality. Therefore we find

$$\mathbb{E}_{T(\ell)}[X_\rho | X_{T(\ell)} = 1] - \mathbb{E}_{T(\ell)}[X_\rho] \leq \mathbb{E}_{T(\ell)}[X_\rho | X_{T(\ell)} = 1] - \mathbb{E}_{T(\ell-1)}[X_\rho] = \mathbb{E}_{T(\ell-1),B}[X_\rho] - \mathbb{E}_{T(\ell-1)}[X_\rho]$$
$$\leq \mathbb{E}_{T(\ell-1),MB'}[X_\rho] - \mathbb{E}_{T(\ell-1)}[X_\rho]$$
$$\leq M(\mathbb{E}_{T(\ell-1),B'}[X_\rho] - \mathbb{E}_{T(\ell-1)}[X_\rho]) = M(\mathbb{E}_{T(\ell)}[X_\rho] - \mathbb{E}_{T(\ell-1)}[X_\rho])$$

where the last two inequalities were by Griffith's inequality[5] (using that $B \leq MB'$) and by concavity of the root marginal w.r.t. the external field along the line from 0 to $B'$, which follows[6] from the fact that the marginal at the root can be computed via the BP recursion and that this recursion is a composition of concave and monotone functions due to Lemma 3.1, hence itself concave.

Summing the corresponding inequality for levels $k = 1$ to $\ell$ we find

$$\ell(\mathbb{E}_{T(\ell)}[X_\rho | X_{T(\ell)} = 1] - \mathbb{E}_{T(\ell)}[X_\rho]) \leq \sum_{k=1}^{\ell}(\mathbb{E}_{T(k)}[X_\rho | X_{T(k)} = 1] - \mathbb{E}_{T(k)}[X_\rho]) \leq M$$

which gives the result. $\qquad\square$

We are now ready to prove the main theorem, Theorem 1.3. For the reader's convenience we recall the statement below; note that we rewrote using $\Phi^*_{Bethe}(\nu^*) = \Phi_{Bethe}(P^*)$ by Theorem 3.4 and the definition of $\Phi^*_{Bethe}$ (see Appendix D).

**Theorem F.2** (Restatement of Theorem 1.3). *Initializing $\nu_{i\to j}^{(0)} = 1$ for all $i \sim j$ and performing $t$ steps of the BP iteration on a graph with $m$ edges we have*

$$0 \leq \Phi^*_{Bethe}(\nu^*) - \Phi^*_{Bethe}(\nu^{(t)}) \leq \sqrt{\frac{8mn(1 + \|J\|_\infty)}{t}}$$

*Proof.* Observe the lower bound is immediate from Lemma 3.5 and the definition of $\nu^*$ as the limit of the iterates from $\nu^{(0)}$. It remains to prove the upper bound.

Let $B > 0$ be arbitrary and define $\nu^*(B)$ to be the optimal BP fixed point when the external field everywhere is increased by $B$. Then since $\nu^*(B)$ corresponds to a global maximum of the Bethe free energy with added external field, we see from the definition of the Bethe free energy that

$$\Phi^*_{Bethe}(\nu^*) + \sum_i B\mathbb{E}_{\nu_*}[X_i] \leq \Phi^*_{Bethe}(\nu^*(B)) + \sum_i B\mathbb{E}_{\nu_*(B)}[X_i].$$

so

$$\Phi^*_{Bethe}(\nu^*) \leq \Phi^*_{Bethe}(\nu^*(B)) + Bn.$$

Define $\nu^{(t)}(B)$ to be the result of the BP iteration after $t$ steps from $\nu^{(0)}$ after having increased the external field at every node by $B$. Observe by monotonicity that $\nu^{(t)} \leq \nu^{(t)}(B)$.

Now we appeal to Lemma 3.2 and Lemma F.1 to see that

$$\Phi^*(\nu^*(B)) - \Phi^*(\nu^{(t)}(B)) \leq \|\nu^{(t)}(B) - \nu^*(B)\|_1 \leq 2m(1 + \|J\|_\infty)/Bt$$

(using that $\nu^{(t)}(B)$ is sandwiched between the output of BP initialized from 0 and from all-1 with additional external field $B$ to get the last inequality from Lemma F.1) hence

$$\Phi^*(\nu^*) - \Phi^*(\nu^{(t)}(B)) \leq Bn + 2m(1 + \|J\|_\infty)/Bt.$$

By Lemma 3.5 this implies that

$$\Phi^*(\nu^*) - \Phi^*(\nu^{(t)}) \leq Bn + 2m(1 + \|J\|_\infty)/Bt$$

as well since $\nu^{(t)} \leq \nu^{(t)}(B) \leq \nu^{(0)}$. Finally we optimize the choice of $B$: solving $Bn = 2m/Bt$ we find $B^2 = 2m(1+\|J\|_\infty)/nt$ so the final upper bound on the excess error is $2\sqrt{2mn(1 + \|J\|_\infty)/t}$. $\qquad\square$

## G   Some Examples

The previous analysis shows how to compute the Bethe free energy by using a small number of rounds of BP to find approximate maximizer of $\Phi^*_{Bethe}$ (on $S$). However, these messages may be far in *parameter space* (e.g. $\ell_1$-distance) from the optimal BP fixed point due to flat directions in the objective. In fact, simple examples show that the number of iterations to reach $o(n)$ distance in parameter space may be *exponential* in $\beta$. These examples also show lower bounds on how quickly the BP estimate for the free energy can converge.

**Example G.1.** Consider the Ising model on the cycle with fixed edge weight $\beta$. Starting from the all-1s initialization, the messages output at time $t$ are all equal to $\tanh(\beta)^t$, so they converge to 0 as $t \to \infty$. Since $1 - \tanh(\beta) = \frac{2e^{-\beta}}{e^\beta + e^{-\beta}} = O(e^{-2\beta})$ we see it takes $\Omega(e^{2\beta})$ iterations for the messages to go below $1/2$.

We also see the that

$$\Phi^*_{Bethe}(\nu^{(t)}) = n\log\left[(1 + \tanh(\beta)^{t+1})^2 + (1 - \tanh(\beta)^{t+1})^2\right] + n\log\frac{e^\beta + e^{-\beta}}{2} - n[\log(1 + \tanh(\beta)^{2t+1})]$$

$$= n\log\left[\frac{2 + 2\tanh(\beta)^{2t+2}}{1 + \tanh(\beta)^{2t+1}}\right] + n\log\frac{e^\beta + e^{-\beta}}{2}$$

$$= n\log(2) + n\log\frac{e^\beta + e^{-\beta}}{2} + n\log(1 + \frac{\tanh(\beta)^{2t+2} - \tanh(\beta)^{2t+1}}{1 + \tanh(\beta)^{2t+1}})$$

$$= n\log(2) + n\log\frac{e^\beta + e^{-\beta}}{2} + \Theta(ne^{-2\beta}\tanh(\beta)^{2t+1}).$$

Therefore if we want to achieve $\epsilon n$ error in $\Phi^*_{Bethe}$ for $\epsilon < e^{-2\beta}$, at least $\Omega(e^\beta)$ iterations of BP are required.

(a) Estimated free energy vs. iteration      (b) Log iterations vs. log residual error for BP

Figure 1: Results of mean-field iteration and BP from all-ones initialization on a $40 \times 40$ square grid with edge weights $\beta = \tanh^{-1}(1/3) \approx 0.347$, with zero external field except for external field of strength 5 at the bottom-left node. In (b) we plot the difference between the true Bethe free energy and the estimated free energy of BP at each iteration on a log-log plot.

**Example G.2.** Consider a $d$-regular graph on $n$ nodes with fixed edge weight $\beta$ and no external field. Let $\beta$ be the critical value given by solving $(d-1)\tanh(\beta) = 1$. Then using symmetry to reduce to a 1-dimensional recursion as before, we see that the BP iteration behaves locally like $x \mapsto x - cx^3$ near the fixed point $x = 0$, where $c = c(d) > 0$. Solving this recurrence, we see that BP converges in parameter space (in $\ell_\infty$ norm) at a $\Theta(1/\sqrt{t})$ rate and the objective value (i.e. estimate for the Bethe free energy density) converges at a $\Theta(1/t^2)$ rate asymptotically.

Furthermore for fixed $\beta$, it's impossible for the convergence rate in parameter space under $\ell_\infty$ norm to be dimension-independent, whereas when $h > 0$ we knew this was indeed true by Lemma F.1:

**Example G.3.** Consider the Ising model on the 2-ary tree for $\beta$ sufficiently large (past the percolation threshold), so the expectation of the root under all-1s is bounded away from 0 by a constant. If the depth of the tree is $k = \Theta(\log n)$, then after $k-1$ rounds of BP initialized from $\vec{1}$ the message from the root to its immediate children will be bounded below by a constant, but after $k$ rounds it will be 0.

Finally, to illustrate the behavior of BP highlighted by our results, we ran the mean-field iteration and BP on a simple $40 \times 40$ square grid example with external field at a single node; the results are shown in Figure 1. As shown, a small number of iterations already suffices to get a good estimate of the mean-field and Bethe free energies; as shown on the log-log plot (Figure 1 (b)), the convergence rate is consistent with a power law decay as shown in Theorem 1.3, although with a better exponent than the worst-case bound shows. This is expected as we expect this model to behave similarly to Example G.2; we chose $\beta$ based on the critical value for the 4-regular tree. In this example, the mean-field iteration converged even faster; again this is consistent with what one would guess based on the behavior in Example 2.5, where one observes that away from the critical $\beta$ the mean field iteration converges faster, at an exponential rate asymptotically.

We also see in Figure 2 the importance of initializing from all-ones; the model is the same as before except that $\beta$ is larger, so that long-range behavior can affect BP. In simple examples like this, BP and mean-field iteration will require at least on the order of the diameter many steps in order to converge if started from all-zeros.

## H    Computing exponentially good BP messages in polynomial time

The bound we proved for BP in Theorem 1.3 showed that if we want to achieve $\epsilon n$ error then $poly(1/\epsilon)$ steps of BP suffice. What if $\epsilon$ is exponentially small? Can a small number of steps of BP achieve $\epsilon$ error? It turns out the answer to this is negative. In Example G.1 from the previous section (Ising model on a line at inverse temperature $\beta$), we saw that BP cannot achieve error $O(e^{-2\beta})$ approximating the free energy without taking at least $\Omega(e^\beta)$ many steps. Therefore, if we want to estimate $\Phi_{BP}^*$ within an exponentially small error in polynomial time we must use a different algorithm.

Figure 2: Comparison of BP and mean-field iteration at all-ones initializes (MF,BP) vs. all-zeros initialization (MF-0,BP-0). The instance is the same as in Figure 1 except that $\beta \approx 0.384$; we see that all-ones initialization leads to quick convergence (consistence with Theorems 1.2 and 1.3) whereas with all-zeros it does not.

In this section we use the ideas developed while analyzing BP to give such algorithm, with runtime $poly(\log(1/\epsilon), n)$ . In order to achieve asymptotically fast convergence to the optimum, we use tools from convex optimization instead of message-passing. Recall that

$$S_{post} = \{\nu \geq 0 : \phi(\nu)_{i \to j} \geq \nu_{i \to j}\}$$

is a convex set (with an obvious separation oracle) and observe that by Theorem 3.4, the optimum $\nu^*$ is the maximizer of the following convex program:

$$\nu^* = \arg\max_{\nu \in S_{post}} \sum_{i,j} \nu_{i \to j}. \tag{11}$$

We show how to compute the maximizer using the ellipsoid method (although a wide variety of methods are applicable, see e.g. [5]). First we analyze the case where $h$ is bounded below, and we show the dependence on $h_{min}$ is very benign.

**Lemma H.1.** *Suppose that $h_{min} := \min h_i > 0$. Then given $\epsilon > 0$, the ellipsoid method applied to* (11) *computes $\nu \in S'$ such that*

$$\|\nu^* - \nu\|_1 \leq \epsilon$$

*after $O(m^2(\log(n/h_{min}) + \log(1/\epsilon)))$ steps of the ellipsoid method.*

*Proof.* Recall from Theorem 2.4 of [5] that the feasible set $S$ contains a ball of radius $r$ and is contained in a ball of radius $R$, then for any convex function $f : \mathbb{R}^d \to [-B, B]$ and $x_t$ the result of $t$ steps of ellipsoid method, satisfies

$$\max_{x \in S} f(x) - f(x_t) \leq \frac{2BR}{r} e^{-t/2d^2}$$

as long as $t \geq 2d^2 \log(R/r)$. Note that our function of interest $\sum_{i,j} \nu_{i \to j}$ is bounded with $B = 2m$ and is contained in $[0,1]^n \subset \mathcal{B}(0, 2\sqrt{n})$. By assumption we see that $S_{post}$ contains $[0, \tanh(h_{min})]^{2m}$ so it contains a ball of radius $\frac{\tanh(h_{min})}{2}$. $\qquad\square$

In order to guarantee the optimization problem is well-behaved, we perturb it by a tiny amount, and this gives an algorithm for the general case. (If we do not perturb the model by adding a tiny external field, $S_{post}$ may be a lower-dimensional, measure-zero set which would be problematic for the ellipsoid method.)

**Theorem H.2.** *Suppose $\epsilon > 0$. There is an algorithm which runs in time $poly(n, \log(1/\epsilon))$ and returns $\nu$ such that*

$$|\Phi^*_{Bethe}(\nu^*) - \Phi^*_{Bethe}(\nu)| \leq \epsilon$$

*Proof.* Fix $B > 0$ to be optimized later. We add external field $B$ everywhere in the model and apply Lemma H.1 to see that we can compute $\nu$ such that

$$\|\nu^*(B) - \nu\|_1 \le \epsilon/2$$

in time $poly(\log(1/\epsilon), n, \log(1/B))$. Then we see by the same argument as in Theorem 1.3 that

$$|\Phi^*_{Bethe}(\nu^*) - \Phi^*_{Bethe}(\nu)| \le Bm + \epsilon/2.$$

Finally taking $B = \epsilon/2m$ shows the result. $\qquad\square$

The same approach works for the mean-field problem as well:

**Theorem H.3.** *Suppose $\epsilon > 0$. There is an algorithm which runs in time $poly(n, \log(1/\epsilon))$ and returns $x$ such that*

$$\Phi_{MF}(x) - \Phi_{MF}(x^*) \le \epsilon$$

*Proof.* Recall the definition of the convex set $S_{pre} := \{x \ge 0 : \tanh^{\otimes n}(Jx + h) \le x\}$ and consider the optimization problem

$$\max_{x \in S_{pre}} \sum_i x_i.$$

Then we do everything the same way as in the proof of Theorem H.2: the algorithm proceeds perturbing the problem by adding a tiny external field $B = \epsilon/2$, and then solving it with ellipsoid method. $\qquad\square$

## Footnotes

[5]This states the root marginal is monotone in the external fields. As with concavity, this can be seen on the tree by writing the root marginal in terms of the BP recursion and using Lemma 3.1.

[6]Alternatively, this can be proved from the GHS inequality as in [7].