[Reviews · NeurIPS 2019]

Reviewer 1



The paper is reasonably well written, but all proofs and technical details are left in the appendix. Some higher-level ideas are provided tho. I did not get to check the proofs. One question that I have is that the theorems make reference to global optimizers, but in the discussion, it appears that global optimizers are those in a restricted domain, namely in [0,1)^n. See lines 208 and 286. So, the question is whether these points are actually global optimizers or optimizers in the specified domain.

Reviewer 2



The proofs are complex and mostly given in the supplementary material. Though I understand the main ideas behind the proofs and find them technically correct, I admit that checking them in detail would take me many days and I did not do it. I have only minor comments: - 66: "a natural to" is a typo, a word is missing - 117: "linear time algorithm" could be confused with linear convergence rate, which is nto the case here. Please, be more precise perhaps (we can epsilon-approximate the functional in time O(1/epsilon). - In eqn (3), the variables x of the MF functional take values {-1,+1}. But e.g Lemma 2.1 speaks about critical points on (0,1]^n. Did I miss something? Please explain. - Section 3.1: Is the "dual" Bethe free energy really derived already in [34]? I did not find it there, please perhaps refer to a concrete place in [34]. This (or similar) "pseudo-dual" to Bethe free energy is proposed also in [T.Werner. Primal View on Belief Propagation. UAI 2010]. - Perhaps, section 2.3 could be omitted. This result is known and not needed for the rest of the paper. POST REBUTTAL: The rebuttal and the other reviews did not change my view on the paper, which I consider very good.

Reviewer 3



The major contributions of this paper are that it proves the global convergence of BP(Theorem 1.3) and VI(Theorem 1.2) on ferromagnetic Ising model with a specific initialization, i.e., to initialize variables to be 1. The proof of Theorem 1.2 is based on the fact that the mean-field free energy function, i.e., \Phi(x) is concave on the set S obtained by the update rule, and then we can use Holder’s inequality to expand the \Phi(x*) - Phi(x_t) and get the upper bounds. The proof of Theorem 1.3 is based on the fact that the norm of \Phi(v)’s gradient is less than 1(Lemma 3.2), and the properties of variable \mu sandwiched between v^0 and final v^T(Lemma 3.5 and Lemma F.1). Other minor contributions include that it provides examples to empirically show the convergence(appendix G) and it shows how to use ellipsoid method to optimize the beliefs(appendix H). I have to admit that I am not familiar with this area, so can only go through a part of the proof, and I am not able to evaluate the originality and quality of this work. Pros: 1. The paper is well organized and the proof is solid. 2. The proof shows that initialization is important to the convergence of BP and VI, which is a very interesting discovery. 3. The proof of BP’s global convergence is important. I think it should be helpful to future works on proving convergence of BP on more general MRFs. Cons: There are some minor problems. 1. I did not see the definition of \phi(v)_{i \to j} in the main paper. I think it is important so should be moved from the appendix to the main paper. 2. I can kind of understand the difference among x_t, x_i, and x_{t,i}, but I think the definitions are not clear enough, and it is useful to use a better way to define them, e.g., x^{(t)}, x_i. 3. In line 575 of appendix, I don’t quite understand the definition of T(v,i,j). That is, what’s the difference between v_{i \to j} and v^{(t)}_{i \to j} 4. In line 622 of the appendix, I don’t quite understand how to get the inequality. In line 626 of the appendix, it is hard for me to derive the inequalities via Lemma 3.2 and F.1, so I think it would be better to add more details of this proof. ======================================== I have read the authors' responses, and other reviewers' comments. I did not change my score. The main reason is that I am not quite familiar with this area, so cannot evaluate the significance of this work. I do not want to overestimate the value of this work.

[Author Response · NeurIPS 2019]

We thank the reviewers for their valuable feedback, comments, and suggestions. Below we respond to the questions raised by the reviewers:

**Reviewer 1:**

1. Our theorems do indeed show convergence of mean-field iteration and belief propagation to global optima over the *entire domain* $[-1, 1]^n$, not just a subset. The reason $[0, 1)^n$ comes up is because we the global optimizer over the entire domain always lies in this orthant (for BP, this is nontrivial but proved as part of Theorem 3.4, for MF it follows straighforwardly by a sign-flipping argument, as $(|x_1|, \ldots, |x_n|)$ always has better objective value than $(x_1, \ldots, x_n)$ in a ferromagnetic model). We will emphasize that global optimum always means the true global optimum in the entire domain in the revision.

**Reviewer 2:**

1. Critical points in $[0, 1)^n$: same response as to reviewer 1 — the global optima over the entire domain $[-1, 1]^n$ always lie in the orthant $[0, 1)^n$, we will make this more explicit in the revised version.

2. Dual Bethe free energy: we agree that this formula did not appear explicitly in Yedidia, Freeman and Weiss and thank the reviewer for the suggested citation. It also appears explicitly in the book by Mezard and Montanari so we will add that as an explicit citation as well.

3. Linear time algorithm, other writing comments: we will make the phrasing more explicit as suggested by the reviewer.

**Reviewer 3:**

1. Line 575: here $\nu$ is anything such that $\nu^{(T)} \leq \nu \leq \nu^{(0)}$ where $\nu^{(t)}$ is defined by BP iteration from $\nu^{(0)}$. We will change $\nu$ to a different letter to minimize confusion in the next revision.

2. Line 622: we apologize for the lack of clarity in this step; in this inequality it's implicit that $h = B$ and we will change all of the occurrences of $h$ to $B$ in the revised version. Then the inequality is immediate from the fact that $\nu^*(B)$ is the maximizer of the Bethe free energy, and $\nu^*$ is one of the feasible points for this maximization. We will add this and show more steps on line 626 for the benefit of future readers.

3. Other notation comments: we will make suggested improvements.

[Meta-Review · NeurIPS 2019]

The reviewers liked the results on convergence of belief propagation algorithms for Ising models under certain settings. As a presentational suggestion, they suggest providing more extensive proof sketches in the main section of the paper.